# FinTech and Sustainable Development: Evidence from China Based on P2P Data

**Xiang Deng, Zhi Huang and Xiang Cheng *** 

School of Economics, Sichuan University, No. 24 South Section 1 Yihuan Road, Chengdu 610064, China; dengxiang@scu.edu.cn (X.D.); 2017321010017@stu.scu.edu.cn (Z.H.)

\* Correspondence: chengxiang@stu.scu.edu.cn

**Abstract:** In the current international context, the ways in which financial technology (FinTech) affects sustainable development need to be urgently identified. However, relevant studies are rare and there is no consensus on the optimal indicator system for sustainable development. Therefore, this study proposes an indicator system to evaluate sustainability and conducts in-depth analysis of the relationship between FinTech and sustainable development based on data of peer-to-peer platforms (P2P) in 31 Chinese provinces. The empirical results show the existence of a U-shaped relationship between FinTech and sustainable development, mainly determined by the pattern of extensive economic growth. Furthermore, heterogeneity analysis indicates that there are significant regional differences in its impact on sustainable development, being significant in China's eastern and central regions and insignificant in the western region; moreover, the impact on the central region is significantly higher than that on the eastern region. Our research not only has strong practical significance but also contributes significantly to the literature on FinTech and sustainable development.

**Keywords:** financial technology; sustainable development; economic growth; social development; consumption emission; environmental governance; peer-to-peer

## 1. Introduction

Given the rapid development of the global economy, environmental problems such as pollution, resource depletion, and ecological imbalances, have risen in scale to become global economic and political challenges for both human survival and development. Sustainable development has gradually become a universal consensus and an important strategic choice for countries worldwide [1,2]. Sustainable development, defined by the World Commission on Environment and Development as "development that meets the needs of the present without compromising the ability of future generations to meet their own needs" [3], involves economic, social, resource, and environmental sustainability [4]. In other words, sustainable development is based on the unification of economic, social, and environmental benefits, and is a comprehensive reflection of all economic, social, environmental, and resource aspects [5].

Since the 2008 global financial crisis, the integration and innovation of emerging technologies and finance haves promoted the development of financial technology (FinTech) [6]. However, as a technology-based financial innovation, FinTech is different from traditional financial innovations [7]. FinTech can be described as a deeper type of innovation or the most cutting-edge technological innovation set in the financial field [8], and includes cryptocurrencies and blockchain, new digital advisory and trading systems, artificial intelligence and machine learning, peer-to-peer lending (P2P), equity crowdfunding, and mobile payment systems [9]. Moreover, FinTech is fundamentally disruptive owing to its major innovations of the financial system and other infrastructure [10], which also affects many aspects of economy, society, and energy.

There is general consensus in academia that FinTech can significantly affect economic growth [11]. However, whether FinTech promotes or constrains economic growth has not been established. This is because, on one hand, FinTech related to communications and data processing can increase the efficiency of financial services by improving financial products and service processes [12,13] and can further promote technological progress for future economic growth [14,15]. In addition, Sun et al. [16] confirm that FinTech can be integrated into new industries and social organizations, resulting in an innovative economic paradigm that could accelerate economic growth. On the other hand, Philippon [9] considers that neither inefficient/low-efficient FinTech nor ineffective/over-regulation is conducive to economic growth. Additionally, Silva [17] confirms that the FinTech landscape may affect the transmission of monetary policy and the effectiveness of macroprudential policy measures in smoothing the financial cycle, thereby constraining economic growth.

Evidence also shows that FinTech has significant effects on social and environmental ecological benefits [18]. In terms of social development, scholars such as [19–21] have confirmed that FinTech's main benefit is its ability to construct a more just and equitable society. However, the potential risks resulting from the development of FinTech cannot be ignored [22–24]. As for environmental ecological development, FinTech can accelerate the deployment of funds for energy and environment projects, promote the construction of renewable energy and environmental infrastructure, and lead to environmental and ecological development by providing cheap and adequate financing [25].

The studies referred to above conduct in-depth analysis of FinTech from different perspectives, finding that FinTech has both promoting and restraining effects on economic growth and social development, and that it can adequately provide financing services for energy and environmental projects. The economy, society, and environment are all important components of sustainable development. The question is whether FinTech impacts on sustainable development. There is a gap in direct research on how FinTech affects sustainable development and its specific direction of action. As a result, a series of research questions have not yet been systematically analyzed or tested. These answers would have significance for both China and the world to seize FinTech development opportunities and a follow-up policy orientation for sustainable development strategies. Therefore, this study constructs a set of evaluation indicators for sustainable development, based on which we analyze the relationship between FinTech and sustainable development. Finally, we further analyze the heterogeneity effects of FinTech and find a U-shaped relationship between FinTech and sustainable development. China's long-term extensive pattern of economic growth is an important reason for this U-shaped relationship. Heterogeneity analysis further indicates that FinTech has significantly different regional impacts on sustainable development, being most prominent in the eastern and central regions of China and insignificant in the western region, while its impact on the central region is highest.

Our study makes three main contributions to the literature. First, given the lack of consensus on a scientific indicator system of sustainable development, we build a sustainable development indicator system for China and use principal component analysis (PCA) to measure the level of sustainable development. Second, this study creatively combines technology-driven financial innovation with sustainable development, to analyze the impact of FinTech on sustainable development, and fills the gaps in current research in this area. Finally, given the fierce competition in FinTech worldwide, our results could help countries to seize FinTech development opportunities and inform the relevant follow-up policy orientation of sustainable development.

The rest of our paper is organized as follows. Section 2 proposes a sustainable development capability index system and calculates the sustainable development level of China. Section 3 constructs the econometric model and explains the regression data and variables. Section 4 analyzes the impact of FinTech on sustainable development. Section 5 conducts further heterogeneity analysis based on Section 4. Section 6 concludes and presents the implications of the research.

## 2. Sustainability Evaluation Criteria

Based on the China Sustainable Development Indicator System (CSDIS; China International Economic Exchange Center and the Earth Institute of Columbia University, 2017), we construct an evaluation system for sustainable development based on four dimensions (economic growth, social development, consumption emissions, and environmental governance). Then, we use PCA to calculate the principal component scores for the four dimensions and a comprehensive score of the sustainable development level. Finally, we describe sustainable development levels in the different regions of China.

### 2.1. Sustainable Development Indicator System

There are numerous international studies on the establishment and comparison of sustainable development indicator systems [26,27]. However, there is still no consensus on any one scientific indicator system among scientific and political communities. In 2017, China Center for International Economic Exchanges and Columbia University's Earth Research Institute [28] announced a scientific indicator system to measure China's sustainable development, the CSDIS. This system requires improvement as follows.

First, consumption is one of the three drivers of economic growth, but its effect on economic growth is not described in the CSDIS. Second, it does not describe the impact of income inequality and urban and rural living environments on social development. Third, the consumption of agricultural resources is not included in consumption emissions. Finally, the completion of investment in industrial pollution control is not considered. Therefore, based on the CSDIS, we constructed an evaluation system for sustainable development based on four dimensions, namely, economic growth, social development, consumption emissions, and environmental governance.

As measurement for economic growth ($f1$), we selected the growth rate of GDP ($X_1$), the tertiary sector's share in GDP ($X_2$), R&D expenditure ($X_3$), urban registered unemployment rate ($X_4$), and proportion of retail sales of social consumption goods in GDP ($X_5$). The growth rate of GDP ($X_1$) directly reflects economic growth [29,30]. The tertiary industry's share in the GDP ($X_2$) reflects the development of the service industry, which has an irreplaceable role in economic growth [31]. R&D expenditure ($X_3$) is regarded as an economic condition created by the government for independent innovation, which reflects the scientific, technological, and economic strengths of a country or region [32]. The change in the unemployment rate ($X_4$) can reflect the utilization of the labor force over a certain period and is an important indicator of the degree of economic prosperity [33]. The ratio of retail sales of social consumption goods in GDP ($X_5$) reflects the level of consumption, demand for which can reflect the effects of purchasing power on economic growth [34–36].

For the assessment of social development ($f2$), we considered government expenditure on education per 10,000 RMB ($X_6$), beds in health institutions per 1000 people ($X_7$), per capita social security and employment finance expenditure ($X_8$), urban–rural income ratio ($X_9$), and proportion of the population residing in urban areas ($X_{10}$). $X_6$–$X_8$ reflect social education, medical care, and the social security system and employment as the main factors affecting social development and stability [37,38]. The urban–rural income ratio ($X_9$) reflects the income gap between urban and rural areas, and is related not only to the interests of each individual but also related to the development of society [39,40]. The proportion of the population residing in urban areas ($X_{10}$) to a certain extent reflects people's living conditions which in rural areas are generally poor with relatively scarce, resources to support healthy living [41]. Therefore, the transfer of rural populations to non-agricultural industries has significantly improved the income and living conditions of farmers [42].

Regarding consumption emissions ($f3$), we used electricity consumption per 10,000 RMB of GDP ($X_{11}$), water consumption per 10,000 RMB of GDP ($X_{12}$), $SO_2$ emissions per 10,000 RMB of industrial output ($X_{13}$), wastewater discharge per unit of industrial output ($X_{14}$), and annual decrease of per capita cultivated land area ($X_{15}$). $X_{11}$ and $X_{12}$ measure electricity and water consumption in GDP growth, respectively, while $X_{13}$ and $X_{14}$ measure emissions of $SO_2$ and sewage in industrial production,

respectively [42,43]. The annual decrease of per capita cultivated land area ($X_{15}$) directly reflects the consumption of cultivated land resources in the process of agricultural development [40].

Finally, as environmental governance ($f4$) indicators, we consider the proportion of government spending on energy conservation and environmental protection in GDP ($X_{16}$), comprehensive utilization rate of industrial solid waste ($X_{17}$), harmless disposal rate of household garbage ($X_{18}$), proportion of green cover in the built-up urban area ($X_{19}$), and investment in the treatment of industrial pollution ($X_{20}$). $X_{16}$ measures the strength of the government's implementation of environmental governance functions [44], while $X_{17}$ and $X_{18}$ measure the treatment of industrial fixed emissions and domestic waste, respectively [45]. $X_{19}$ reflects the ecological greening of the city [39] and $X_{20}$ measures the progress of industrial pollution policies [46].

Since the secondary indicators within the four dimensions of economic growth, social development, consumption emissions, and environmental governance are of different types and have regional differences, it was necessary to use reliability analysis to examine their internal consistencies to determine whether they can measure the corresponding primary indicators. To employ PCA, the negative secondary indicators were uniformly converted into positive ones and then standardized. Cronbach's alpha yields results above 0.65, which means that all internal indicators were consistently highly reliable and could be used as representative indicators of the primary indicators.

Table 1 provides an overview of the proposed sustainable development indicator system, the sources of the indicators, and the results of the reliability test.

**Table 1.** Sustainable development indicator system.

| First-Level Indicator | Secondary Indicators | Source of Indicators | Cronbach's Alpha |
|---|---|---|---|
| Economic Growth ($f1$) | X1: Growth rate of GDP (%) <br> X2: Proportion of tertiary sector to GDP (%) <br> X3: R&D expenditure (10,000 RMB) <br> X4: Urban registered unemployment rate (%) <br> X5: Proportion of retail sales of social consumption goods in GDP (%) | Lin [30], Lavoie [29], CSDIS <br> Flynn et al. [31], CSDIS <br> Inekwe [32], CSDIS <br><br> Fanti & Gori [33], CSDIS <br><br> Fisher & Hof [34], Balios et al. [35], Sun et al. [36] | 0.6825 |
| Social Development ($f2$) | X6: Government expenditure on education (100 million RMB) <br> X7: Beds in health institutions per 1000 people <br> X8: Per capita social security and employment finance expenditure (RMB) <br> X9: Urban–rural income ratio <br> X10: Proportion of population residing in urban areas (%) | Dempsey et al. [37], Murayama et al. [38], CSDIS <br><br><br><br> Li et al. [39], Strezov et al. [40] Hartig & Kahn [41], Haseeb et al. [42] | 0.7099 |
| Consumption Emissions ($f3$) | X11: Electricity consumption per 10,000 RMB of GDP (kw·h) <br> X12: Water consumption per 10,000 RMB of GDP ($m^3$) <br> X13: $SO_2$ emissions per 10,000 RMB of industrial output (kg) <br> X14: Sewage discharge per unit of industrial output (t) <br> X15: Annual decrease of per capita cultivated land area ($m^2$) | Kanemoto et al. [43], Haseeb et al. [42], CSDIS <br><br><br><br><br> Strezov et al [40] | 0.7539 |
| Environmental Governance ($f4$) | X16: Proportion of government spending on energy conservation and environmental protection in GDP (%) <br> X17: Comprehensive utilization rate of industrial solid waste (%) <br> X18: Harmless disposal rate of household garbage (%) <br> X19: Proportion of green cover in the built-up urban area (%) <br> X20: Completed industrial pollution control investment (10,000 RMB) | Wang et al. [44], CSDIS <br><br> Rametsteiner et al. [45], CSDIS <br><br> Li et al. [39] <br><br> Feng et al. [46] | 0.7112 |

## 2.2. Principal Component Analysis of Sustainable Development

To measure the level of sustainable development in China, we conducted PCA based on Section 2.1. To observe the degree of correlation between variables, the Kaiser–Meyer–Olkin (KMO) test is carried out on the four data dimensions. All test results were greater than 0.6, indicating that PCA could play a role in data reduction. The following steps were undertaken.

**Step 1**: Extract the principal components of economic growth, social development, energy consumption, and environmental governance.

According to the extraction criterion of the eigenvalue being greater than or equal to 1, the extracted principal component cumulative variance contribution rate was below 80%. For the principal component and the original secondary index to have similar ability to interpret the primary index, the principal component cumulative variance contribution was obtained. The criterion with a rate greater than 80% extracted the principal components and selected three principal components in each dimension (see Table 2).

**Table 2.** Principal components and variance contribution rates of the four dimensions.

| Comp. | Eigenvalue | Proportion | Cumulative | Comp. | Eigenvalue | Proportion | Cumulative |
|---|---|---|---|---|---|---|---|
| | Economic Growth | | | | Consumption Emissions | | |
| 1 | 2.227 | 0.445 | 0.445 | 1 | 2.251 | 0.450 | 0.450 |
| 2 | 1.020 | 0.204 | 0.649 | 2 | 1.173 | 0.235 | 0.685 |
| 3 | 0.816 | 0.163 | 0.812 | 3 | 0.862 | 0.685 | 0.857 |
| | Social Development | | | | Environmental Governance | | |
| 1 | 2.395 | 0.479 | 0.479 | 1 | 2.538 | 0.508 | 0.508 |
| 2 | 1.180 | 0.236 | 0.715 | 2 | 0.756 | 0.151 | 0.659 |
| 3 | 0.727 | 0.145 | 0.860 | 3 | 0.745 | 0.149 | 0.808 |

Note: Comp. is the abbreviation of component and the tables below are the same.

**Step 2**: Obtain the eigenvectors of the principal components.

Table 3 shows the eigenvectors of the three principal components of economic growth, social development, energy consumption, and environmental governance.

**Table 3.** Eigenvectors of principal components for the four dimensions.

| Indicator | Comp. 1 | Comp. 2 | Comp. 3 | Indicator | Comp. 1 | Comp. 2 | Comp. 3 |
|---|---|---|---|---|---|---|---|
| | Economic Growth | | | | Consumption Emissions | | |
| X1 | −0.5146 | 0.2540 | 0.3122 | X11 | 0.4932 | −0.5113 | 0.0176 |
| X2 | 0.4862 | 0.4035 | −0.2328 | X12 | 0.5211 | −0.3589 | −0.3776 |
| X3 | 0.4652 | −0.4991 | −0.2414 | X13 | 0.5094 | 0.3500 | −0.0350 |
| X4 | 0.3616 | −0.2644 | 0.8696 | X14 | 0.2946 | 0.6786 | −0.3725 |
| X5 | 0.3894 | 0.6736 | 0.1842 | X15 | 0.3728 | 0.1636 | 0.8468 |
| | Social Development | | | | Environmental Governance | | |
| X6 | 0.5216 | −0.2229 | −0.4020 | X11 | −0.5012 | 0.1889 | 0.3296 |
| X7 | 0.2941 | 0.7504 | 0.0880 | X12 | 0.4093 | 0.6983 | 0.3122 |
| X8 | 0.4922 | 0.2739 | 0.4777 | X13 | 0.4683 | −0.3286 | −0.4392 |
| X9 | 0.3560 | −0.5529 | 0.5920 | X14 | 0.4693 | 0.3296 | −0.0986 |
| X10 | 0.5220 | −0.0812 | −0.5020 | X15 | 0.3764 | −0.5100 | 0.7689 |

**Step 3**: Calculate the main component scores for the four dimensions.

The principal component scores for the four dimensions were calculated in conjunction with the eigenvectors in Table 3 for economic growth, social development, energy consumption, and environmental governance. Based on the comprehensive evaluation function $fi = \sum_{j=1}^{3} a_{ij} f_{ij}$, we calculated the comprehensive scores of the four first-level indicators. Here, for $i = 1, 2, 3, 4,$

*fi* denotes the principal component scores of the four dimensions, $f_{ij}$ is the score of principal components *j* of first-level indicator *i*, and *a* is the variance contribution rate.

　　　**Step 4**: Calculate comprehensive score of the sustainable development level.

　　　Based on the scores of the four dimensions calculated in Step 3, PCA of sustainable development was conducted. According to the principle that the cumulative variance contribution rate is above 80%, three principal components were selected, and the comprehensive score of sustainable development was calculated according to the corresponding characteristic vector. A statistical description of the results is given in Section 2.3.

　　　**Step 5**: End.

### 2.3. Statistical Description of Sustainable Development Levels by Region

　　　Table 4 reports the sustainable development of China's provinces in 2009 and 2017. The sustainable development levels are represented by *esgc*. Compared with 2009, China's sustainable development level had improved significantly by 2017, as the number of provinces with a sustainable development level score greater than 2 increased from 1 to 5, while the number of scores greater than 0 increased from 3 to 29. At the same time, the gap in sustainable development levels between provinces was still large. In 2017, the sustainable development levels of most provinces were concentrated in the interval [0, 1), the highest level being that of Beijing, followed by Guangdong and Jiangsu. The lowest was that of Ningxia, followed by Guizhou.

**Table 4.** Provincial sustainable development levels.

| Type | 2009 | | 2017 | |
|---|---|---|---|---|
| | No. | Provinces (Descending) | No. | Provinces (Descending) |
| Type 1 ($esgc \geq 2$) | 1 | Beijing | 5 | Beijing, Guangdong, Jiangsu, Zhejiang, Shandong |
| Type 2 ($1 \leq esgc < 2$) | 1 | Guangdong | 8 | Gansu, Hainan, Hubei, Shanghai, Guangxi, Henan, Tianjin, Liaoning |
| Type 3 ($0 \leq esgc < 1$) | 1 | Zhejiang | 16 | Anhui, Tibet, Xinjiang, Shanxi, Inner Mongolia, Chongqing, Jilin, Qinghai, Yunnan, Hebei, Heilongjiang, Hunan, Shanxi, Sichuan, Jiangxi, Fujian |
| Type 4 ($esgc < 0$) | 28 | Shanghai, Jiangsu, Tibet, Gansu, Shandong, Hainan, Shanxi, Guizhou, Tianjin, Fujian, Xinjiang, Liaoning, Jiangxi, Henan, Qinghai, Hebei, Guangxi, Hubei, Jilin, Hunan, Anhui, Heilongjiang, Shanxi, Chongqing, Yunnan, Ningxia, Inner Mongolia, Sichuan | 2 | Guizhou, Ningxia |

　　　For analysis of the regional differences in sustainability levels further, Table 5 presents the average scores and dynamics of the sustainable development levels in the eastern, central, and western regions. The results show that the average level of sustainable development continued to increase in all three regions. The sustainable development level in the eastern region increased the most from 2009 to 2017, with a score increase of 2.1789, followed by the central region (2.1009) and the western region (1.909). There were obvious regional differences in the levels of sustainable development. In 2017, the average level of sustainable development in the eastern region was highest, at 2.0928, followed by the central region at 0.6866 and the western region at 0.4899, forming a step structure.

**Table 5.** Scores and dynamic changes in sustainable development levels in the eastern, central, and western regions.

| Region/Year | 2009 | 2011 | 2013 | 2014 | 2015 | 2016 | 2017 |
|---|---|---|---|---|---|---|---|
| Eastern | −0.0861 | 0.5520 | 1.1022 | 1.4482 | 1.6862 | 1.8821 | 2.0928 |
| Central | −1.4143 | −1.3987 | −0.6509 | −0.1650 | 0.2299 | 0.4672 | 0.6866 |
| Western | −1.4191 | −1.3443 | −0.6773 | −0.2754 | −0.0242 | 0.1815 | 0.4899 |

## 3. Model and Data

Based on the analysis in Section 2, the impact of FinTech is shown to involve important aspects of sustainable development, such as the economy, society, and the environment; however, the direct relationship between FinTech and sustainable development needs to be tested empirically. To this end, we use the fixed effect (FE) model to test this relationship based on P2P (peer-to-peer lending) platform data. Furthermore, the dynamic system generalized method of moments (DS-GMM), expansion of sample capacity, and replacement of explanatory variables were used to test the robustness of the results.

### 3.1. Econometric Model

To analyze the effect of FinTech on sustainable development further, we built an econometric model based on [47,48], as follows:

$$esgc_{it} = \beta_0 + \beta_1 gfin_{it} + \beta_2 gfin_{it}^2 + \beta_i controls_{it} + \alpha_i + \gamma_t + \varepsilon_{it} \tag{1}$$

where $esgc_{it}$ denotes the level of sustainable development in year $t$ of province $i$, $\beta_0$ is the intercept term, $\alpha_i$ is the individual effect, $\gamma_t$ is the time fixed effect, $\varepsilon_{it}$ is the random error term, $gfin_{it}$ is the level of normal P2P platforms, $gfin_{it}$ and $gfin_{it}^2$ are the main explanatory variables, $controls_{it}$ is the set of control variables, $\beta_1$ and $\beta_2$ are the coefficients on the explanatory variables, and $\beta_i$ denotes the coefficient vector on the control variables.

To tackle the heteroscedasticity and skewness of variables and thereby avoid the influence of outliers, we winsorized the main control variables at 5%. In terms of model selection, for the panel data of the listed companies, the Hausman test was performed on all benchmark regressions. The results reject the null hypothesis of the random effects model. As such, we used the FE model for benchmark testing with Equation (1). To overcome the problems of autocorrelation and heteroscedasticity, we performed a double clustering adjustment on the individual and time of the sample, and then we estimated the model by adding control variables [49]. Furthermore, to ensure the reliability of the results, based on benchmark regression, DS-GMM, expansion of the sample capacity, and replacement of explanatory variables were used to test result robustness.

### 3.2. Main Variables and Data Description

### 3.2.1. Explained Variable

We took the sustainability level of each province in China ($esgc_{it}$) as the explained variable, based on Section 2. The level of sustainable development in our study is characterized by four dimensions: economic growth ($f1$), social development ($f2$), consumption emissions ($f3$), and environmental governance ($f4$). Owing to the large number of indicators for evaluating sustainable development capability, multiple collinearity issues arise among the internal indicators of the four dimensions. As per Section 2.2, PCA was used to calculate the total score of the sustainable development ability in each province, which was then used as a proxy variable for the sustainable development level.

### 3.2.2. Explanatory Variables

The explanatory variable $gfin_{it}$ denotes the level of FinTech development. Its value is represented by the number of normal P2P platforms, and the total number of P2P platforms ($fin_{it}$) was used for robustness testing. We used the number of normal P2P platforms as the proxy variable of FinTech ($gfin_{it}$) for the following reasons.

First, according to the classification of the Basel Committee, P2P belongs to the "deposit-and-loan and capital raising class," and is an important type of FinTech [8]. Second, FinTech is dominated by traditional financial institutions in China, is still in its infancy, whereas P2P began to develop as early as 2006 [50]. Third, FinTech is developing rapidly in China, but the relevant statistical mechanisms are missing, with only a few P2P platforms having adequate data availability. Therefore, we represent the development level of FinTech by the number of P2P platforms.

### 3.2.3. Control Variables

This study controls for a range of variables that may influence the level of sustainable development in a region. First, we considered the impact of national policies, industrial development, and local lending on sustainability. Fiscal and monetary policies are important tools for the government to regulate macroeconomic growth and social development [51]. Therefore, in this study we controlled the impact of fiscal expenditure ($gspd_{it}$) and monetary policy ($mp_t$) on the level of sustainable development, $mp_t$ is calculated according to Lu and Yang [52] using Equation (2):

$$mp = rm2 - ggdp - gcpi \tag{2}$$

where $rm2$ denotes the growth rate of the money supply, $ggdp$ is the GDP growth rate, and $gcpi$ is the growth rate of the consumer price index.

Furthermore, industry exhibits significant energy consumption and pollution discharge, indicating that an increase in industrial output value is accompanied by increases in energy consumption and pollution discharge, which affects the sustainable development of resources and the environment [53]. Therefore, we included industrial output value ($sgdp_{it}$) as a control variable. Moreover, as capital has a significant impact on economic growth and the society and loans are an important source of capital, we measured the scale of loans ($loan_{it}$) as the number of loans in each province and controlled for its impact on sustainable development ability.

Second, this study considered the impact of the population age structure, population distribution, and educational level of the labor force. The old-age dependency rate ($oldrate_{it}$) can reflect the aging of the population in a region. The higher this rate, the less human capital is available for production, which is not conducive to sustainable development [54]. Population density ($rpeople_{it}$) is an important indicator of population distribution and regional environmental pressure. Population-intensive growth to promote economic growth generates more domestic waste, increases resource consumption, raises environmental treatment difficulties [55]. The education level of the labor force ($hedu_{it}$) is an important manifestation of the level of human capital, which is, in turn, a key determinant of a region's sustainable development [56]. Therefore, according to Mincer and Polachek [57], the educational years of illiterate, elementary, junior high school, high school, junior college, undergraduate, and postgraduate students were set to 0, 6, 9, 12, 15, 16, and 19 years, respectively. $hedu_{it}$ equals to the average number of years of education in the labor force and was used as a control variable.

Third, we considered regional market dynamics, regulatory difficulties, and the impact of imports and exports on sustainability. The number of enterprises is used as a "barometer" for optimizing the business environment and stimulating market vitality. It is also an important embodiment of the economic development potential of a region and an important reflection of the degree of social development and the intensity of resource consumption [58]. We adopted the number of regional corporate legal entities ($enprs_{it}$) as a proxy of regional market vitality and as a control variable. Regional regulatory information asymmetry can affect regulatory efficiency and affect the resource

environment. As such, to control for the influence of regulatory information asymmetry, we used regulatory information asymmetry ($inforasy_{it}$) as a control variable. Most previous studies have used physical distance to measure information asymmetry [59]. By considering the impact of the number of regional microfinance companies on regulatory information and avoiding endogeneity, we used the ratio of "land area to small loan companies" in provinces and cities to measure the asymmetry of regulatory information. Industrialized countries are increasingly dependent on material and energy resources from other regions of the world for production and consumption, and transfer environmental burden abroad through imports, thereby extending their responsibility for environmental impacts and social consequences from the national to the global level [60]. Therefore, we controlled for the impact of imports and exports ($openness_{it}$) on sustainable development.

Finally, we used time variable $T$ and its square $T^2$. As the sample begins in 2009, time was considered as the year minus 2008. Table 6 shows the descriptive statistics for the main regression variables.

**Table 6.** Descriptive statistics.

| Variable | Obs. | Mean | SD | 25% | 50% | 75% |
|---|---|---|---|---|---|---|
| $gfin_{it}$ | 200 | 54.14 | 106.144 | 4.5 | 16 | 46 |
| $fin_{it}$ | 200 | 76.76 | 157.904 | 5 | 20 | 63 |
| $gspd_{it}$ | 279 | 35.364 | 18.414 | 20.596 | 33.067 | 46.195 |
| $mp_t$ | 279 | 4.467 | 5.564 | 2.6 | 3.2 | 5 |
| $sgdp_{it}$ | 279 | 0.803 | 0.716 | 0.325 | 0.607 | 1.082 |
| $loan_{it}$ | 279 | 2.394 | 2.026 | 0.953 | 1.748 | 3.259 |
| $oldrate_{it}$ | 279 | 13.016 | 2.862 | 10.9 | 12.85 | 14.8 |
| $rpeople_{it}$ | 279 | 680.773 | 123.710 | 271.986 | 535.588 | |
| $hedu_{it}$ | 279 | 9.608 | 0.990 | 9.027 | 9.905 | 10.093 |
| $enprs_{it}$ | 279 | 33.412 | 34.340 | 11.398 | 23.023 | 42.793 |
| $inforasy_{it}$ | 248 | 1.988 | 13.542 | 0.043 | 0.077 | 0.182 |
| $openness_{it}$ | 279 | 0.18 | 0.200 | 0.014 | 0.033 | 0.109 |

Note: Obs. denotes the number of observations and the tables below are the same.

### 3.3. Data Sources

We used P2P platform data over 2009–2017 for China's 31 provinces as the research sample. The data period is chosen because FinTech's proxy variable (number of P2P platforms) is available for as early as 2009, while the latest data for other relevant variables are available up to 2017. Data on the number of P2P platforms came from Zero-One Finance, and the other data came from the China Statistical Yearbook, China Population & Employment Statistics Yearbook, China Labour Statistical Yearbook, People's Bank of China, and Wind Database.

## 4. Empirical Analysis

We used the FE model to test the relationship between FinTech and sustainable development. Then, we employ mediation effect analysis (MEA) based on the benchmark test results. To ensure the robustness of the results, the DS-GMM, expansion of sample capacity, and replacement of the explanatory variables, were used. Finally, the variable coefficient FE model was utilized to analyze the heterogeneity effects of FinTech.

### 4.1. Benchmark Test Results

Table 7 reports the benchmark estimated results based on data for 2011-2017. The results show that, in the case of controlling other variables, the coefficient on $gfin_{it}$ was significantly negative at the 5% significance level, and the coefficient on $gfin_{it}^2$ was significantly positive at the 10% significance level. Thus, the results indicate a U-shaped relationship between FinTech and sustainable development.

When FinTech is less than the critical value, it has a restraining effect on sustainable development, and once it exceeds this value, it promotes sustainable development.

**Table 7.** FinTech's impact on sustainability.

| Variable | (1) | (2) | (3) | (4) | (5) | (6) |
|---|---|---|---|---|---|---|
| | $esgc_{it}$ | $esgc_{it}$ | $esgc_{it}$ | $esgc_{it}$ | $esgc_{it}$ | $esgc_{it}$ |
| $gfin_{it}$ | −0.00278** | −0.00339*** | −0.00343*** | −0.00135 | −0.00233** | −0.00303** |
| | (0.00116) | (0.00119) | (0.00113) | (0.00106) | (0.00106) | (0.00131) |
| $gfin_{it}^2$ | 0.0280 | 0.0367** | 0.0349** | 0.0111 | 0.0281* | 0.0419* |
| | (0.0185) | (0.0179) | (0.0168) | (0.0149) | (0.0154) | (0.0220) |
| $gspd_{it}$ | 0.00205 | −0.000202 | 0.00207 | 0.00233 | 0.00792 | 0.00624 |
| | (0.0115) | (0.0111) | (0.0116) | (0.0115) | (0.0112) | (0.0108) |
| $gspd_{it-1}$ | 0.00967 | 0.0147 | 0.0106 | 0.0147 | 0.0195* | 0.0238** |
| | (0.00999) | (0.0107) | (0.0112) | (0.0110) | (0.0113) | (0.0112) |
| $mp_t$ | | 0.0128 | 0.00978 | 0.00927 | 0.00113 | 0.00909 |
| | | (0.0259) | (0.0261) | (0.0246) | (0.0247) | (0.0297) |
| $mp_{t-1}$ | | −0.0125 | −0.0112 | −0.0146 | −0.0168 | −0.0118 |
| | | (0.0167) | (0.0165) | (0.0155) | (0.0149) | (0.0184) |
| $mp_{t-2}$ | | 0.0212** | 0.0240*** | 0.0168** | 0.0183** | 0.0188** |
| | | (0.00831) | (0.00824) | (0.00848) | (0.00806) | (0.00847) |
| $sgdp_{it}$ | | | 0.283 | 0.0186 | −0.574* | −0.689** |
| | | | (0.235) | (0.256) | (0.333) | (0.337) |
| $loan_{it}$ | | | −0.0229** | −0.0179* | −0.0140 | −0.0162 |
| | | | (0.0114) | (0.00986) | (0.0105) | (0.0106) |
| $oldrate_{it}$ | | | | -0.0270 | −0.0538* | −0.0579* |
| | | | | (0.0321) | (0.0306) | (0.0325) |
| $rpeople_{it}$ | | | | −0.00943*** | −0.00855*** | −0.00808*** |
| | | | | (0.00234) | (0.00223) | (0.00233) |
| $hedu_{it}$ | | | | | 0.412*** | 0.433*** |
| | | | | | (0.118) | (0.113) |
| $enprs_{it}$ | | | | | 0.00741*** | 0.00715** |
| | | | | | (0.00277) | (0.00284) |
| $inforasy_{it}$ | | | | | | 0.212 |
| | | | | | | (0.148) |
| $openness_{it}$ | | | | | | 2.246 |
| | | | | | | (1.952) |
| $T$ | 0.704*** | 0.924*** | 0.975*** | 0.839*** | 0.892*** | 0.870*** |
| | (0.131) | (0.217) | (0.223) | (0.217) | (0.217) | (0.234) |
| $T^2$ | −0.0339*** | −0.0502*** | −0.0553*** | −0.0439** | −0.0543*** | −0.0523*** |
| | (0.0105) | (0.0163) | (0.0171) | (0.0173) | (0.0174) | (0.0175) |
| Constant | −2.873*** | −3.713*** | −3.979*** | 1.436 | −2.724 | −3.414* |
| | (0.475) | (0.688) | (0.758) | (1.606) | (1.957) | (2.035) |
| Obs. | 183 | 183 | 183 | 183 | 183 | 183 |
| Number of pid | 31 | 31 | 31 | 31 | 31 | 31 |
| Adj. $R^2$ | 0.695 | 0.697 | 0.700 | 0.717 | 0.731 | 0.734 |

Note: ***, **, and * denote significance at the 1%, 5%, and 10% levels, respectively, while double clustering robust standard errors are in parentheses. Pid denotes the provinces of China and the tables below are the same. In addition to the dynamic panel regression, both individual and vintage fixed effects were included.

In addition, the regression coefficients on the control variables are consistent with those in previous studies. Specifically, the regression coefficients on $gspd_{it-1}$ and $mp_{t-2}$ were significantly positive at 5%, indicating that fiscal and monetary policies have a lagging effect on sustainable development, while expansionary fiscal and monetary policies have a positive effect on sustainable development. Fiscal policy directly affects the socio-economic structure through transfer payments, government purchases, etc., thereby promoting the growth of national economic output and affecting the level of

environmental governance. Meanwhile, monetary policy indirectly regulates the economic structure and social environment by affecting the currency market. Therefore, monetary policy has a stronger lag [61]. Regardless of whether they are expansionary, fiscal or monetary policies can provide funds for economic and social development, promote innovation and environmental regulation, and improve the level of sustainable development [62]. Industrial development is a significant constraint to sustainable development. Over the past 2 decades, China's economy has been driven mainly by factor investment, and the environmental challenges brought about by industrial development are becoming critical constraints to China's sustainable development [63]. The regression coefficients on $oldrate_{it}$ and $rpeople_{it}$ pose significant constraints on sustainable development, consistent with existing research findings [64]. There is a significant positive correlation between $enprs_{it}$ and sustainable development. Based on data from the China Statistical Yearbook, the proportion of legal entities in the tertiary sector continued to increase during 2009–2017, from 67.87% to 70.91%, thereby upgrading the industrial structure and promoting sustainable development [65]. The coefficient on $T$ is significantly positive and that on $T^2$ is significantly negative, with a downward trend between 2016 and 2017. The sustainability levels of the 31 provinces over 2009–2016 show a clear upward trend but, in 2017, more than one-fourth of the provinces declined compared with 2016. Furthermore, $loan_{it}$, $inforasy_{it}$, and $openness_{it}$ are not significant for sustainable development.

### 4.2. Additional Analysis Based on Benchmark Test Results

Table 7 shows that FinTech and sustainability have a U-shaped relationship, as FinTech constrains sustainable development when it is less than a critical value and promotes sustainable development once a threshold is exceeded. We find that FinTech has both positive and negative effects on economic growth. The benchmark results indicate that regional industrial added value poses a significant constraint to sustainable development, showing characteristics of extensive growth. Therefore, we aim to determine whether the U-shaped relationship between FinTech and sustainable development is related to China's pattern of economic growth. To test this effect, we used MEA for further analysis based on the economic growth rate ($ggdp_{it}$), which we select instead of economic growth ($f1$) measured in Section 2 for the following two reasons. First, the GDP growth rate is the most representative indicator, reflecting the economic growth trend of a country or region. Second, $f1$ is a comprehensive indicator covering economic growth, industrial growth, social consumption capacity, economic driving force, employment status, etc., but the analysis of China's economic growth pattern does not involve social consumption capacity, economic driving force, employment status, etc. Column (1) of Table 8 reports the estimated results for FinTech and $ggdp_{it}$. Column (2) shows the regression result of FinTech and consumption emissions ($f3$). Column (4) shows the regression result of FinTech and environmental governance ($f4$). Columns (3) and (5) present the regression results of increasing $ggdp_{it}$.

**Table 8.** Results of mediation effect analysis.

| Variables | (1) | (2) | (3) | (4) | (5) |
|---|---|---|---|---|---|
| | $ggdp_{it}$ | $f3_{it}$ | $f3_{it}$ | $f4_{it}$ | $f4_{it}$ |
| $gfin_{it}$ | 0.00946*** | −0.00145** | −0.000701 | −0.00159** | −0.000792* |
| | (0.00329) | (0.000635) | (0.000429) | (0.000676) | (0.000455) |
| $gfin_{it}^2$ | −0.0962* | 0.0202* | 0.0126* | 0.0220* | 0.0139* |
| | (0.0517) | (0.0107) | (0.00726) | (0.0114) | (0.00774) |
| $sgdp_{it}$ | 3.032* | −0.298* | −0.0567 | −0.338** | −0.0806 |
| | (1.790) | (0.162) | (0.168) | (0.171) | (0.179) |
| $ggdp_{it}$ | | | −0.0795*** | | −0.0849*** |
| | | | (0.0296) | | (0.0316) |
| Controls | YES | YES | YES | YES | YES |
| Obs. | 183 | 183 | 183 | 183 | 183 |
| Number of pid | 31 | 31 | 31 | 31 | 31 |
| Adj. $R^2$ | 0.761 | 0.730 | 0.777 | 0.721 | 0.770 |

Note: ***, **, and * denote significance at the 1%, 5%, and 10% levels, respectively, while double clustering robust standard errors are in parentheses. To unify the measurement direction of the four dimensions of sustainable development, the measurement of resource consumption is symbolized. Therefore, the opposite result should be considered when interpreting economic meaning. Due to space limitations, the regression results for the control variables are available on request.

The results in Column (1) of Table 8 show that FinTech has an inverted U-shaped relationship with economic growth, while Columns (2) and (4) show that FinTech's has a total effect on consumption emissions ($f3$) and environmental governance ($f4$). Combined with the regression results of Equations (1), (3), and (5), $ggdp_{it}$ is found to significantly weaken the influence of FinTech on $f3$ and $f4$ (the coefficient became smaller and its significance decreased). These results indicate that the mediation effect of $ggdp_{it}$. FinTech can influence consumption emissions and environmental governance by affecting economic growth, whereby China's long-term extensive pattern of economic growth is an important reason for the U-shaped relationship between FinTech and sustainable development. Therefore, influenced by FinTech's innovation, the economic growth rate first increased and then decreased, while the level of sustainable development showed first a downward and then a rising trend.

While FinTech has an inverted U-shaped relationship with economic growth, as a technology-driven financial innovation, FinTech has demonstrated that it cannot match the advantages of traditional financial innovation in terms of increasing financial demand, reducing financial service costs, improving financial efficiency, and effectively driving economic growth [11,14,66]. However, as FinTech has expanded in scale, the financial landscape has changed, financial risks have gradually accumulated, and inefficient innovation driven by rent-seeking and commercial theft has increased [9,17,67]. In addition, FinTech's traditional regulatory systems and regulations based on prudential, functional, and behavioral supervision cannot effectively deal with the status quo of de-intermediation and decentralized financial transactions. As a result, ineffective supervision or over-regulation inhibits the effects of FinTech, leading to a lack of financial support in the real economy and constraining economic growth [10].

### 4.3. Robustness Tests

Benchmark regression analysis was used to estimate the model by adding control variables and confirms the robustness of the regression results to some extent. To ascertain the reliability of the results, the DS-GMM, expansion of sample capacity, and replacement of explanatory variables are employed as robustness tests. Columns (1) and (2) of Table 9 show the DS-GMM regression results, while Columns (3) and (4) present the results of the expansion of the sample capacity and replacement of explanatory variables, respectively.

### 4.3.1. System GMM Estimation

In addition to the variables mentioned in this section, many other factors influence sustainable development, including past level of sustainable development. Therefore, we built a dynamic panel econometric model for robustness testing:

$$esgc_{it} = \beta_0 + \beta_1 esgc_{it-1} + \beta_2 esgc_{it-2} + \beta_3 gfin_{it} + \beta_4 gfin_{it}^2 + \beta_i controls_{it} + \alpha_i + \gamma_t + \varepsilon_{it} \qquad (3)$$

where $esgc_{it-1}$ and $esgc_{it-2}$ denote the levels of sustainable development in years $t-1$ and $t-2$, respectively, of province $i$; the other variables are as per Equation (1). The model was estimated using the DS-GMM, and the explained lag periods were selected as control variables. Column (1) of Table 9 reports the two-stage estimation results. According to the Sargan test results, the hypothesis that all instrumental variables are valid could not be rejected. Column (2) of Table 9 reports the estimation results after considering heteroscedasticity and sequence correlation and testing the sequence correlation of the disturbance term. The analysis results show that the disturbance term had no autocorrelation. The DS-GMM results show that the coefficient on $gfin_{it}$ was significantly negative and that on $gfin_{it}^2$ was significantly positive; furthermore, there was a U-shaped relationship between FinTech and sustainable development, which is consistent with the benchmark regression results.

### 4.3.2. Expansion of Sample Capacity

To test the robustness of the benchmark results further, 2009 and 2010 data are added to expand the sample capacity to 2009–2017. The regression results in Column (3) of Table 9 show that $gfin_{it}$ was significantly negative at the 5% level and $gfin_{it}^2$ was significantly positive at the 10% significance level, indicating a U-shaped relationship between FinTech and sustainable development levels, which is consistent with the benchmark regression results.

### 4.3.3. Replacement of Explanatory Variables

We used the total number of P2P platforms ($fin_{it}$) to replace ($gfin_{it}$) as an explanatory variable for robustness testing. The regression results in Column (4) of Table 9 show that the coefficient on $fin_{it}$ was significantly negative and the coefficient on $fin_{it}^2$ was significantly positive at the 5% level, indicating a U-shaped relationship between FinTech and sustainable development levels; these regression results are consistent with the benchmark regression results.

In summary, the robustness results of the DS-GMM, expansion of sample capacity, and replacement of explanatory variables are consistent with the benchmark regression results, indicating that the benchmark regression results are stable and confirming a U-shaped relationship between FinTech and sustainable development.

**Table 9.** Robustness tests.

| Variable | (1) | (2) | (3) | (4) |
|---|---|---|---|---|
|  | $esgc_{it}$ | $esgc_{it}$ | $esgc_{it}$ | $esgc_{it}$ |
| $esgc_{it-1}$ | 0.809*** | 0.737*** |  |  |
|  | (0.122) | (0.138) |  |  |
| $esgc_{it-2}$ | −0.166 | −0.164 |  |  |
|  | (0.102) | (0.168) |  |  |
| $gfin_{it}$ | −0.00288*** | −0.00295** | −0.00303** |  |
|  | (0.000540) | (0.00141) | (0.00131) |  |
| $gfin_{it}^2$ | 0.0432*** | 0.0470* | 0.0419* |  |
|  | (0.00974) | (0.0261) | (0.0220) |  |
| $fin_{it}$ |  |  |  | −0.00169** |
|  |  |  |  | (0.000695) |
| $fin_{it}^2$ |  |  |  | 0.0150** |
|  |  |  |  | (0.00688) |
| Controls | YES | YES | YES | YES |
| Obs. | 183 | 185 | 183 | 184 |
| Number of pid | 31 | 31 | 31 | 31 |
| Adj. $R^2$ | - | - | 0.734 | 0.732 |

Note: ***, **, and * denote significance at the 1%, 5%, and 10% levels, respectively, while double clustering robust standard errors are in parentheses. Due to space limitations, the regression results for the control variables are available on request.

## 5. Heterogeneity Effects of FinTech

China's vast territory, resource endowments, geographical conditions, policies, and regulations have generated serious regional development imbalances, with the development differences between the eastern, central, and western regions being particularly significant. These differences may lead to regional differences in the effects of FinTech on sustainability. Therefore, we used variation coefficient FE analysis to verify this difference.

Column (1) of Table 10 reports the results of FinTech's impact on sustainable development in the eastern, central, and western regions. The results show a U-shaped relationship between FinTech and sustainable development. The relationships in the eastern and central regions are significant, but insignificant in the western region. According to the results of Table 8, the extensive pattern of economic growth is an important reason for the U-shaped relationship. For further analysis of reasons for FinTech's regionally heterogeneous effects on sustainability, we estimated the respective effects of FinTech on economic growth, consumption emissions, and environmental governance. In Table 10, Columns (2) to (4) suggest that, there is an inverted U-shaped relationship between FinTech and both economic growth and consumption emissions, but a significant U-shaped relationship between FinTech and environmental governance. These results are significant in the eastern and central regions, but insignificant in the western region.

Possible reasons are the remarkable regional differences in China's FinTech development level [68]. Superior regional advantages and strong economic strength in the eastern region have created favorable conditions for FinTech development [36]. Meanwhile, the eastern region has the country's first pilot region aimed at promoting the integration of technology and finance, leadings to a higher level of FinTech [69,70]. The central region is adjacent to the developed region, and thus, is easily affected by innovation diffusion form the eastern region; it has the second-highest level of FinTech development [71]. However, most parts of the western regions are far from China's developed coastal areas. Economic weakness, insufficient innovation capacity, and low demand for FinTech products all lead to the western region's low FinTech level [70]. Based on the results, FinTech significantly impact on the economic growth of the eastern and central regions, while the impact on the western region is not significant. According to Table 9, the impact of FinTech on economic growth will further affect consumption emissions and environmental governance, and the results, presented in Columns (3) and (4) of Table 10

are significant in the eastern and central regions, but insignificant in the western region. Taking into account economic growth, consumption emissions, and environmental governance, FinTech significantly impacts on the level of sustainable development in the eastern and central regions, but not in the western.

In addition, the impact of FinTech on the central region is significantly greater than that on the eastern region. One plausible explanation is that the relatively high level of financial development of eastern region, has constrained the impact of FinTech on economic growth, yielding a comparatively gentle impact [36]. However, the central region has a relatively low degree of financial development, and the economic development has led to huge demand for FinTech products [72]. Therefore, once FinTech is successfully implemented, the economic growth rate in the western region will rise rapidly, but with the emergence of inefficient innovation and ineffective supervision, the accumulation of financial risks will further suppress the rate of economic growth. Therefore, under the extensive economic growth pattern, the impact of FinTech on sustainable development in the central region is significantly greater than that in the eastern region, which is indicate by the more rapid rate of change in the sustainable development level of the central region than that of the eastern region.

**Table 10.** Regression results for FinTech.

| Variable | (1) | (2) | (3) | (4) |
|---|---|---|---|---|
| | $esgc_{it}$ | $ggdp_{it}$ | $f3_{it}$ | $f4_{it}$ |
| $gfin_{it}$ | −0.00349** | 0.0100*** | −0.00168** | −0.00183** |
| (East) | (0.00148) | (0.00357) | (0.000718) | (0.000765) |
| $gfin_{it}$ | −0.0107 | 0.0273 | −0.00521 | −0.00549 |
| (Central) | (0.00989) | (0.0193) | (0.00483) | (0.00513) |
| $gfin_{it}$ | 0.00178 | −0.0339 | 0.000887 | 0.00102 |
| (West) | (0.00693) | (0.0235) | (0.00341) | (0.00360) |
| $gfin^2_{it}$ | 0.0513** | −0.115** | 0.0248** | 0.0269** |
| (East) | (0.0240) | (0.0551) | (0.0116) | (0.0124) |
| $gfin^2_{it}$ | 1.591* | −2.857* | 0.785* | 0.826* |
| (Central) | (0.871) | (1.672) | (0.426) | (0.452) |
| $gfin^2_{it}$ | −0.912 | 3.146 | −0.454 | −0.488 |
| (West) | (0.672) | (2.653) | (0.331) | (0.350) |
| Controls | Yes | Yes | Yes | Yes |
| Obs. | 183 | 183 | 183 | 183 |
| Number of pid | 31 | 31 | 31 | 31 |
| Adj. $R^2$ | 0.743 | 0.768 | 0.740 | 0.730 |

Note: ***, **, and * denote significance at the 1%, 5%, and 10% levels, respectively, while double clustering robust standard errors are in parentheses. Due to space limitations, the regression results for the control variables are available on request.

## 6. Conclusions and Implications

Our study constructed evaluation indicators for sustainable development and further measured the sustainability levels of different provinces in China. Then, we established an empirical model of FinTech and sustainability levels, and tested their relationship using P2P data. Finally, we considered the heterogeneity of FinTech. We found (1) that there is a U-shaped relationship between FinTech and sustainable development; (2) that the extensive pattern of economic growth is an important reason for the U-shaped relationship between FinTech and sustainable development; and (3) FinTech shows significant regional differences in its impact on sustainable development, being significant in the eastern and central regions of China but insignificant in the western region. Furthermore, its impact in the central region is significantly greater than that in the eastern region.

Given the fierce competition in FinTech worldwide, our results are significant for countries around the world, including China, highlighting the need to seize the development opportunities of FinTech and to provide follow-up policy orientation for sustainable development. On the one hand, sustainable

development not only meets the needs of the present but also benefits future generations. It is a form of development that encompasses economic growth, social development, consumption emissions, and environmental governance in multiple dimensions. Therefore, in the new stage of development, countries should pay more attention to sustainable development instead of focusing on only a single indicator, such as GDP. On the other hand, we should pay more attention to the development of FinTech and its effects. First, for countries around the world, FinTech is an important driver of sustainable development. In the process of development, we should promote the construction of an umbrella regulatory sandbox for FinTech. At the same time, research and development investment in the underlying technologies required by FinTech should be increased to improve innovation efficiency, reduce imitative innovation with low efficiency, and expand high-efficiency FinTech. Second, in the process of FinTech's development, countries should speed up the improvement of its regulations and policies, build a technology-driven supervision system, promoting the breakthrough and reconstruction of financial supervision models and theories, and preventing and resolving the negative effects of ineffective or excessive regulation of FinTech. Third, in the process of promoting sustainable development, to give full play to the positive effect of FinTech on sustainable development, countries must reform extensive patterns of economic growth, strengthen the transformation and upgrading of the industrial structure, and promote sustainable development with low consumption, pollution, and emissions. Finally, to encourage FinTech and implement financial regulations, regional differences in FinTech influence should be fully considered and differentiated policies and measures should be formulated.

Our study has some limitations. For example, in analyzing the impact of FinTech on sustainable development, our study employs P2P data from China and does not involve data from other countries. In future research, we aim to focus on the following two aspects. First, we plan to analyze the impact of FinTech on sustainable development in Organization for Economic Co-operation and Development. Second, we aim to conduct a comparative analysis on the impact of FinTech on sustainable development from developed and developing countries.

**Author Contributions:** X.D. conceived the study; Z.H. was responsible for data curation and drafting the paper; X.C. proposed the method.

**Funding:** This research was funded by the National Natural Science Foundation of China, Grant Nos. 71742004, 71673194, and 71473169 for D.X. and the Graduate Student's Research and Innovation Fund of Sichuan University, Grant No.2018YJSY001 for C.X. The APC was funded by Sichuan University.

**Acknowledgments:** We thank Li Li (School of Economics, Sichuan University) for her valuable comments, suggestions, and support. The authors would also like to thank the editors and anonymous reviewers for their constructive comments and helpful suggestions.

**Conflicts of Interest:** The authors declare no conflict of interest.

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
