# Peer review of "FinTech and Sustainable Development: Evidence from China Based on P2P Data"

_sustainability, doi:10.3390/su11226434_

Round 1

Reviewer 1 Report

The authors presented the up to date topic on the influence of financial technology (FinTech) on sustainable development.

They introduced very interesting methodology of analysis of sustainable development on the regional level as well as  the impact of FinTech on sustainable development.

However, the authors could deepen the results of quantitative research, explaining the obtained relations based on the literature analysis. Namely, there should be enumerated the reasons of heterogeneity effects of FinTech in Chinese regions, e.g.  why the relationship of FinTech on sustainability in the eastern and central regions is significant, while it is  insignificant in the western region? Why the impact of FinTech on the central region is significantly greater than in the eastern region? Another relations worth explanation: FinTech shows an inverted U-shaped relationship for economic growth that is significant in the eastern and central regions and insignificant in the western region; FinTech has a  significant U-shaped relationship with consumption emissions in the eastern and central regions, while the relationship is insignificant in the western region; it also has a significant U-shaped relationship with environmental governance in the eastern and central regions that is not significant in the western region.

The authors conclude that there is a U-shaped relationship between FinTech and sustainable development (lines 472-472). However, the the relationship in the western region is insignificant. It is worth to add the discussion part which should take into account reference of results obtained in a given study to other research occurring in the literature.

Reviewer 2 Report

- Overall this study is executed well. However, I believe the discussion and Implications should be further strengthen based on the sustainable indicator level selected for the study.

Also, the authors should include relevant references from this journal (Sustainability). There is only one paper cited.

On a quick search, I found one below

Al. Othman, F.A.; Sohaib, O. Enhancing Innovative Capability and Sustainability of Saudi Firms. Sustainability 20168, 1229.

Reviewer 3 Report

The comments and suggestions for the authors of this article are in the document attached. 

I wish you good luck with your work!

The Reviewer

Round 2

Reviewer 3 Report

Dear Authors of Manuscript: sustainability-635954,

Please find attached my comments concerning your current version of this Article.

Kind regards,

The Reviewer

This manuscript is a resubmission of an earlier submission. The following is a list of the peer review reports and author responses from that submission.